# Sunlight Photocatalytic Performance of ZnO Nanoparticles Synthesized by Green Chemistry Using Different Botanical Extracts and Zinc Acetate as a Precursor

**DOI:** 10.3390/molecules27010006

**Published:** 2021-12-21

**Authors:** Juan López-López, Armando Tejeda-Ochoa, Ana López-Beltrán, José Herrera-Ramírez, Perla Méndez-Herrera

**Affiliations:** 1Facultad de Ciencias Químico-Biológicas, Universidad Autónoma de Sinaloa, Av. Las Américas S/N, Culiacan 80000, Sinaloa, Mexico; jrll@uas.edu.mx (J.L.-L.); bamoita@uas.edu.mx (A.L.-B.); 2Centro de Investigación en Materiales Avanzados, Laboratorio Nacional de Nanotecnología, Miguel de Cervantes 120, Chihuahua 31136, Chih, Mexico; a.tejeda.ochoa@gmail.com

**Keywords:** zinc oxide, green synthesis, nanoparticles, photocatalysis

## Abstract

In this work, the assessment of *Azadirachta indica*, *Tagetes erecta*, *Chrysanthemum morifolium*, and *Lentinula edodes* extracts as catalysts for the green synthesis of zinc oxide nanoparticles (ZnO NPs) was performed. The photocatalytic properties of ZnO NPs were investigated by the photodegradation of methylene blue (MB) dye under sunlight irradiation. UV-visible (UV-Vis) spectroscopy, Fourier Transform Infrared (FTIR) spectroscopy, Transmission Electron Microscopy (TEM), X-ray Diffraction (XRD), Thermogravimetric (TGA), and Brunauer-Emmett-Teller analysis (BET) were used for the characterization of samples. The XRD results indicate that all synthesized nanoparticles have a hexagonal wurtzite crystalline structure, which was confirmed by TEM. Further, TEM analysis proved the formation of spherical and hemispherical nanoparticles of ZnO with a size in the range of 14–32 nm, which were found in aggregate shape; such a size was well below the size of the particles synthesized with no extract (~43 nm). ZnO NPs produced with *Tagetes erecta* and *Lentinula edodes* showed the best photocatalytic activity, matching with the maximum adsorbed MB molecules (45.41 and 58.73%, respectively). MB was completely degraded in 45 min using *Tagetes erecta* and 120 min using *Lentinula edodes* when subjected to solar irradiation.

## 1. Introduction

Zinc oxide nanoparticles (ZnO NPs) are one of the nanostructured materials of greatest interest due to their wide range of applications in pharmaceutical [1], electronic [2], medical [3], power generation [4], and environmental [5] fields, which is due to their multiple properties such as semiconductors, pyroelectrics, piezoelectrics, catalysts, optoelectronics [6], antimicrobial, and anticancer activity [1].

Zinc oxide is an environmentally friendly and abundant-in-nature semiconductor with n-type conductivity and a wide band gap of 3.3 eV. ZnO has been extensively used in the photocatalytic degradation of organic pollutants owing to its good quantum efficiency, wide band gap, and nontoxic nature [7]. The advantage of this semiconductor is that the ZnO can absorb a significant fraction of the solar spectrum compared with other photocatalysts such as TiO_2_. ZnO absorbs UV light with a wavelength below 387 nm; therefore, its photocatalytic activity is limited to the UV region. Solar light consists of less than 5% UV radiation, which is why it is necessary to improve this efficiency by modifying its band gap.

ZnO NPs can be synthesized by several methods such as physical vapor deposition (PVD) and chemical vapor deposition (CVD). However, these methods usually require high temperature, multiple steps, and sophisticated equipment. Wet chemical processes such as spray pyrolysis, hydrothermal processes, sol–gel processing, precipitation, and coprecipitation are cost-effective, scalable, and have been used in the synthesis of a wide variety of ZnO nanostructures [8].

It is worth mentioning that traditional nanoparticle production uses toxic materials such as solvents and surfactants. Currently, green chemistry for the fabrication of metal oxide nanomaterials has garnered enormous attention as an alternative technique to produce nanomaterials with the assistance of botanical extracts. The use of plant extracts reduces the required amounts of the precursor and other chemicals. Furthermore, it is possible to synthesize metallic and nonmetallic materials as well as modify the size, morphology, area–volume relationship, porosity, and other properties. It is noteworthy that plant extracts are composed of numerous biomolecule compounds, such as polyphenols, flavonoids, carbonyl, and protein compounds. These biomolecule compounds could serve as green capping agents and templates and, accordingly, could play pivotal roles in the fabrication process [9].

The precipitation method has been used extensively to prepare ZnO NPs [9,10,11,12,13,14]. Recently, Lam et al. synthesized iron-doped ZnO (Fe-ZnO) nanoparticles according to a simple and green method utilizing *Hibiscus rosa-sinensis* leaf extract, reporting the properties of these nanoparticles for the inactivation of *Escherichia coli* (*E. coli*) under sunlight irradiation [9]. Another example of this method has been reported with *Gliricidia sepium* leaf extract using direct-sunlight-driven selective photodegradation of methylene blue [12], showing that it is possible to obtain a high percentage of degradation and selectivity of the material with certain types of dyes. Researchers have also studied the impact of ZnO NPs obtained with *Azadirachta indica* on antibacterial and antimicrobial properties; they have found that Gram-positive and Gram-negative bacterial growth decreased when the concentration of ZnO NPs increased [13,14]. Additionally, leaf extracts of *Tagetes erecta*, *Chrysanthemum morifolium*, and *Lentinula edodes* have been used for the synthesis of silver nanoparticles for use in antibacterial activity, whose results have been favorable [15,16,17,18,19].

In this research, ZnO nanoparticles were synthesized using different natural extracts, namely, *Azadirachta indica*, *Tagetes erecta*, *Chrysanthemum morifolium*, and *Lentinula edode**s*. The precursor was obtained by the precipitation method via the reaction between zinc acetate (Zn(CH_3_CO_2_)_2_) and sodium hydroxide (NaOH) in aqueous solutions. The effects of the extract on the morphological and photocatalytic properties of the ZnO products were evaluated.

## 2. Results and Discussion

### 2.1. Characterization of Zinc Oxide Nanoparticles

#### 2.1.1. UV-Visible Spectroscopy

Figure 1a shows the UV-Vis spectra of a suspension of 0.01 g ZnO NPs obtained in the presence of different aqueous botanical extracts, which were dispersed in 100 mL of water. Semiconductor oxide nanoparticles presented a maximum absorption peak in the range of 360–375 nm, which is in agreement with Sharma et al., who reported a UV-Vis absorbance peak between 360 and 375 nm for ZnO NPs [20].

From the absorbance data, the energy band gap (Eg) was estimated using Tauc’s plot method (Figure 1b); the calculated Eg values are 2.99, 3.01, 3.01, 3.06, and 3.18 eV for ZnO NPs with no extract, *Tagetes erecta*, *Lentinula edodes*, *Azadirachta indica*, and *Chrysanthemum morifolium*, respectively. It is well-known that bulk ZnO has a band gap of 3.37 eV [21], and the decrease in the band gap energy found here can be related to the presence of defect states caused by the botanical extracts, as well as by their influence on size control during nanoparticle formation [5]. Several studies have shown the potential use of different phytochemicals (e.g., flavonoids, tannins and polyphenolics, terpenoids, amines and ketones) obtained in botanical extracts to control the size and shape of different types of nanoparticles [22,23].

#### 2.1.2. Fourier Transform Infrared (FTIR) Spectroscopy Analysis

FTIR results for ZnO nanoparticles synthesized with each type of extract can be seen in Figure 2. The observed peaks can be mainly related to phytochemicals present in the aqueous botanical extracts. These compounds are believed to have an important role during nanoparticle formation, helping in the reduction process, controlling the size, giving stability, functionalizing the nanoparticles, modifying the morphology, and thus the properties and applications of the nanostructured material.

The FTIR spectra show absorption bands corresponding to hydrogen-bonded O–H stretching (3400 cm^−1^) [24], CO_2_ interference (2340 cm^−1^) [24], stretching for (–C=C–) bond of alkenes (2160 cm^−1^) [25], primary amines and sulfur compounds (2030 cm^−1^) [26], O–H bending (1580 cm^−1^), C=C stretching mode in aromatic compounds (1420 cm^−1^) [19], and C–O stretching (1030 cm^−1^) [27]. The deep absorption band from 610 cm^−1^ to lower wavenumbers is attributed to wurtzite-type ZnO and the Zn–O stretching band is normally found around 443 cm^−1^ [28]; however, the spectrometer used here is not capable of detecting such wavenumbers.

#### 2.1.3. Transmission Electron Microscopy (TEM) and High-Resolution Transmission Electron Microscopy (HRTEM)

TEM micrographs of ZnO NPs samples are displayed in Figure 3. The nanoparticles have a spherical and hemispherical morphology and their size varies as a function of the extract used. The nanoparticles’ sizes were measured directly during the TEM analysis in some representative micrographs. Table 1 summarizes the variation in nanoparticle size. As can be seen, the smaller nanoparticles are those obtained in the presence of *Azadirachta indica*, while the largest nanoparticles belong to those obtained with no extract.

A more comprehensive TEM analysis was performed on the ZnO NPs aggregates. As an example, Figure 4 presents the case of ZnO NPs synthesized with *Lentinula edodes*. Figure 4a shows an aggregate of ZnO NPs from which the SAED pattern in Figure 4b was acquired. Such a pattern presents Debye–Scherrer rings, meaning that the selected area is polycrystalline, which is due to the random distribution of the nanoparticles. Figure 4c corresponds to a HRTEM image of a ZnO nanoparticle, which was processed through GMSS for obtaining the image in Figure 4d; a *d* spacing of 0.27 nm was measured. The indexing of the SAED and the FFT (inset in Figure 4d) patterns reveal that the nanoparticles are composed mainly of the hexagonal wurtzite phase of ZnO. The results obtained for the ZnO NPs samples synthesized with the other extracts were similar [29].

#### 2.1.4. X-ray Diffraction (XRD)

Figure 5 presents the XRD patterns of the synthesized ZnO NPs. All the indexed peaks correspond to the hexagonal wurtzite phase of ZnO (JCPDS card No. 79-0206) and no other peaks are observed. These results complement and confirm the TEM analysis described above. In addition, the shape of the XRD patterns shows peak broadening in samples synthesized with *Lentinula edodes* and *Azadirachta indica* extracts, being more evident in the latter, while sharp peaks can be found in samples with no extract and *Chrysanthemum morifolium* extract. The peak broadening indicates a reduction in crystallite size, as can be seen in Table 2. The crystallite size of the nanoparticles varies according to the particle size shown in Table 1. The decrease in crystallite size can be related to the capping agent effect of the extract, which helps to control the size and prevent the ZnO NPs from aggregation [30].

#### 2.1.5. Thermogravimetric Analysis (TGA)

Figure 6 depicts the TGA thermograms to identify the thermal transformation of the Zn-complex obtained with the different botanical extracts. The initial weight loss of the samples between 78 and 100 °C was due to the evaporation of water remaining in the ZnO samples. The subsequent weight loss at higher temperatures is attributed to the different volatile phytocomponents present in the samples from the plant and loss of own material. A total weight loss at 800 °C of 3.28, 2.79, 2.14, 1.96, and 0.97% was calculated for samples synthesized with *Azadirachta indica*, *Tagetes erecta*, *Lentinula edodes*, *Chrysanthemum morifolium*, and without extract, respectively.

#### 2.1.6. Nitrogen Adsorption–Desorption (BET) Isotherms

Figure 7 shows the nitrogen adsorption–desorption isotherms for ZnO NPs synthesized with no extract and those with the best photocatalytic performance, as will been seen below in the following section. Surface area (S_BET_), pore volume, and average pore diameter were determined through these isotherms, as seen in Table 3.

The adsorption isotherms of Figure 7 show a type IV structure with H3 hysteresis loops, according to the IUPAC classification [31]. Hysteresis loops observed in 0.6–1.0 (P/P_0_) indicate the presence of mesopores. This behavior indicates that there is a correlation between the shape of the hysteresis loop and the texture of the nanostructures (e.g., pore size distribution, geometry, and connectivity). In this particular case, it can be observed in Table 3 that the nanoparticles obtained without botanical extract present smaller surface area and smaller average pore diameter than those prepared with *Tagetes erecta* or *Lentinula edodes* extracts.

### 2.2. Photocatalytic Activity

#### 2.2.1. Photocatalytic Degradation of Methylene Blue (MB)

Methylene blue (MB) is a heterocyclic aromatic chemical compound that is mainly used as a dye for textiles and paper, as well as in the biomedical industry. MB reaches water bodies through effluent discharges affecting ecosystems and aquatic life, hence the importance of carrying out effective treatments for the degradation of these pollutants. Due to its characteristics, MB was used in this work as a model molecule to test the photocatalytic performance of the synthesized materials.

Figure 8 shows the absorbance spectra plotted as a function of the sunlight exposure time for the MB samples without catalyst and with the catalyst that presented the best catalytic performance (*Tagetes erecta*).

Figure 8a shows the behavior of MB in the absence of a catalyst, which shows photolysis effects since the dye is sensitive to solar radiation [32]. In the presence of the catalyst (Figure 8b), during the time of dark equilibrium, the MB molecules adsorbed on the catalyst in different percentages according to the botanical extract used for ZnO synthesis. The importance of adsorption of pollutant molecules has been reported as a critical step for the oxidation process to be carried out efficiently during photocatalysis; the adsorbed MB on catalyst undergoes a faster degradation than MB molecules unabsorbed [33,34]. The adsorption percentages in the present work were 6.31, 9.37, 31.04, 58.73, and 45.41% for ZnO without extract, *Azadirachta indica*, *Chrysanthemum morifolium*, *Lentinula edodes*, and *Tagetes erecta*, respectively. According to these results, it can be stated that the catalysts that adsorbed the highest amount of MB molecules presented the best photocatalytic performance. It is possible to appreciate in Figure 9 that all catalysts achieved complete photodegradation at different times: 270, 120, 150, 120, and 45 min with *Azadirachta indica*, no extract, *Chrysanthemum morifolium*, *Lentinula edodes,* and *Tagetes erecta*, respectively. Extracts are known to have different effects on the nanoparticles, such as controlling their size, morphology, and porosity, but they also function as capping agents modifying the catalytic properties.

#### 2.2.2. Kinetic Study of Photocatalytic Degradation of Methylene Blue

From the data in Figure 9, plots of ln(*C*_0_/*C_x_*) versus sunlight exposure time were constructed (Figure 10) to evaluate the efficiency of the photocatalytic process through the rate constant *k* of the process. This constant is given by the pseudo-first-order kinetic reaction (Equation (1)), valid for millimolar pollutant concentrations [35].
(1)$$\ln\left(\frac{C_{0}}{C_{x}}\right)=kt$$
where *C*_0_ is the initial methylene blue concentration, *C_x_* is the residual dye concentration in solution after a certain time, *k* is the rate constant (min^−1^), and *t* is the exposure time (min).

The behavior in the plots of Figure 10 is the same as that found in the plots of Figure 9. It is important to note that the calculation of the rate constant has a contribution from the effect of photolysis, and possibly from adsorption, which is why the apparent rate constant (*k_app_*) is the product of (adsorption + photolysis + photocatalysis) processes.

The difference lies in the kinetics (*k_app_*) of each of them, which is associated with the different botanical extracts used during the synthesis process. Table 4 compiles the apparent rate constant obtained from the linear adjustment of the graphs in Figure 10 performed in OriginPro^®^ software. These results indicate that when ZnO NPs are present, the degradation of the dye by photocatalytic effect is increased.

ZnO NPs obtained with *Azadirachta indica* present the slowest degradation kinetics compared with the other extracts. On the other hand, extracts of *Lentinula edodes* and *Tagetes erecta* proved to be faster catalysts (higher reaction rate constant), reaching complete degradation in 120 and 45 min, respectively. Particularly, the extract of *Tagetes erecta* presented the best photocatalytic performance, degrading the dye approximately 3.3 times faster than the catalyst synthesized with no extract. This behavior may be attributed to the porosity and the different functional groups present in the *Tagetes erecta* flower, among which terpenoids, saponins, and tannins stand out [36].

#### 2.2.3. Recyclability of ZnO Nanoparticles

Considering that *Tagetes erecta* extract showed the best photocatalytic performance, it was used to evaluate the stability of ZnO NPs. The catalyst was recovered and tested three times under similar reaction conditions. To reuse the catalyst, ZnO NPs were recovered after centrifugation and subsequently dried at 100 °C for 2 h to remove moisture. Figure 11 illustrates the degradation percentage obtained in each cycle for the same reaction time (45 min) established by Figure 9. As can be seen, the removal percentage decreases slightly as the number of cycles increases (5.5% in the third cycle); so, it can be said that the ZnO catalyst retains appropriate activity (adsorption–photocatalytic), demonstrating that it can be recovered and reused.

According to the overall results, it was found that depending on the composition of the extract used, the size, adsorption capacity, and catalytic properties of the nanostructures can be influenced. The aqueous extracts of neem leaves (*Azadirachta indica*) present compounds of terpenoid- and flavonoid-type (nimbin, azadirachtin, nimbidiol, quercetin, nimbidin), among others [37,38]. In the case of flowers of chrysanthemum (*Chrysanthemum morifolium*), the presence of volatiles, flavonoids, and flavonoid glycosides has been reported, with quercitrin, myricetin, and luteolin-7-glucoside being the abundant flavonoids [39,40]. Concerning shiitake mushroom extracts (*Lentinula edodes*), they contain saccharides (mainly glucooligosaccharides) and phenolic compounds [41]. Finally, *Tagetes erecta*, the extract with the best photocatalytic performance (highest reaction rate), is reported to contain terpenoids, flavonoids, tannins, and saponins (rutin, kaempferol, quercetin, kaempferitrin, and β-sitosterol) [42,43]. The main phytochemical structures for the plants used in this investigation are shown in Figure 12, Figure 13, Figure 14 and Figure 15, which were drawn using the Biovia draw software based on the data from [37,38,39,40,41,42,43]. In general, each extract has a complex profile of compounds, which can vary for the same species depending on the type of solvent used in the extraction, environmental factors, postharvest storage, and processing conditions.

Terpenoids and flavonoids are the compounds in common in the different extracts. It should be noted that the two extracts showing the best performance were reported to contain saccharides, tannins, saponins, and phenolic compounds. These compounds could be responsible for modifying the properties of the materials in such a way that they increase the adsorption of the contaminant molecules and, thus, significantly improve the rate of photodegradation. It has been reported that, depending on the composition of the botanical extract used, a high affinity between the functional groups of the extracts and dye molecules can occur [44]. Further, surfaces modified with functional groups have been tested and it has been demonstrated how the interaction with methylene blue is improved, and therefore, the adsorption is increased [45,46,47,48].

The findings reported in this work indicate that it is possible to design a catalyst material that presents high interaction–adsorption with the pollutant—depending on the chemical characteristics of the pollutant and the functional groups of the extracts to be used—in such a way that MB degradation can be favored.

## 3. Materials and Methods

### 3.1. Reagents and Collection of Botanical Material

The chemical products for the experimental study were all reagent grade and used as received. Zinc acetate (98%) and methanol MeOH (99.87%) were bought from FagaLab. Sodium hydroxide (95%), acetone (99.5%), sodium sulfate (99.75%), potassium chloride (99%), and dimethylsulfoxide (DMSO 99.95%) were purchased from ICR. Ethanol (96°) was acquired from Alfimex and 2-propanol (99.9%) from J.T.Baker.

Neem leaves (*Azadirachta indica*) were gathered from in and around the Faculty of Chemical–Biological Sciences Campus. Flowers of cempasuchil (*Tagetes erecta*) and chrysanthemum (*Chrysanthemum morifolium*), as well as shiitake mushrooms (*Lentinula edodes*) were acquired from a local flower shop and food store. The botanical material was washed with running tap water to remove contamination, followed by a double-distilled water wash, dried in a tray dryer, and stored before its use.

### 3.2. Preparation of Botanical Extracts

A total 50 g of ground neem leaves were heated with 500 mL double-distilled water in a beaker at 60 °C for 30 min until the color of the water was turned to brownish. The extract was collected and cooled at room temperature followed by filtration. Meanwhile, 25 g of ground material from cempasuchil, chrysanthemum, and shiitake mushroom were used. The obtained filtrates were stored under refrigeration at 4 °C for further experiments.

### 3.3. Synthesis of Zinc Oxide Nanoparticles

A total 100 mL of 0.5 M of zinc acetate was mixed with 100 mL of each botanical extract (*Azadirachta indica*, *Tagetes erecta*, *Chrysanthemum morifolium*, *Lentinula edodes*). The zinc-acetate-extract solution was placed in an ultrasonic bath for 15 min, followed by magnetic stirring at high velocity for 10 min more. The solution was mixed with 1 M NaOH solution added drop-by-drop until pH 7 was reached. The resulting solution was heated at 70 °C for 1 h, then filtered to remove the supernatant and washed twice with double-distilled water and ethanol. Once filtered, the precipitate was left overnight in a flask at 80 °C. Finally, the powder was sintered at 400 °C for 1 h. The different powdered forms of ZnO nanoparticles were stored for future studies.

### 3.4. Characterization of Zinc Oxide Nanoparticles

The ultraviolet-visible spectra were acquired by a UV-Vis spectrometer (Velab VE-5100UV); wavelength scanning was performed from 340 to 500 nm. Fourier transform infrared (FTIR) spectroscopy analysis of ZnO nanoparticles was performed to detect the functional groups associated with the synthesized nanoparticles; a FTIR System spectrum GX developed by Perkin Elmer was used, operated in the wavenumber range from 4000 to 440 cm^−1^. A JEOL JEM2200FS + CS high-resolution transmission electron microscope (HRTEM) operated with an accelerating voltage of 200 kV was used to study the morphology of the synthesized ZnO NPs and to obtain the selected area electron diffraction (SAED) patterns. The Miller indices identification on SAED and in the fast Fourier transform (FFT) patterns were made with the Crystallographic Tool Box (CrystBox) software using ring analysis mode (ring GUI) [49]. Image processing was performed with Gatan Microscopy Suite Software (GMSS). N_2_ adsorption–desorption isotherms were obtained by the Brunauer–Emmett–Teller (BET) method in a Quantachrome model Nova 4200 e analyzer, taking 11 points of 0.05 to 0.3 of relative pressure (P/P_0_); from this analysis, the surface area and the pore distribution were determined. Thermogravimetric analysis (TGA) was achieved in a Q600 equipment developed by TA Instruments from ambient temperature till 800 °C with a heating rate of 10 °C/min in air atmosphere; the ZnO nanoparticles were placed into an alumina pan with an initial weight of around 33 mg. Finally, powder X-ray diffraction (XRD) patterns of ZnO nanoparticles were recorded by an X’Pert3 MRD diffractometer developed by Panalytical using Cu-Kα (*λ* = 0.1542 nm) monochromatic radiation; the analysis was performed in the 2*θ* range from 20 to 80° using a step size of 0.05. The average crystallite size (*D*) of the nanoparticles were determined based on XRD peak broadening using the Scherrer method (Equation (2)) [50]:(2)$$D=\frac{0.9\lambda }{\beta cos\left(\theta \right)}$$
where *λ* is the X-ray wavelength, *β* is the full-width at half-maximum intensity (FWHM), and *θ* is the diffraction angle.

### 3.5. Measurement of Photocatalytic Activity

The photocatalytic performance of the prepared powders was evaluated by the degradation of methylene blue under direct sunlight (681–702 W/m^2^ in Culiacán, Sinaloa). The photocatalytic activity of catalysts was tested as follows: first, 0.3 g of each sample was dispersed in 300 mL of 10 mg/l methylene blue solution (*C*_0_), regulating the pH to 10; this value was selected according to previously reported pH optimization results [51]. Then, the solution was placed in the dark with constant stirring for 45 min to achieve equilibrium of methylene blue adsorption on ZnO nanoparticles before being exposed to radiation [32,52]. Subsequently, samples were taken every 30 min (if the kinetics was fast, samples were taken every 5 min) and centrifuged at 8000 rpm to separate the nanoparticles in suspension; the supernatants were evaluated by UV-Vis spectroscopy at 665 nm, whose wavelength corresponds to the most important band (azo dye content). The corresponding removal efficiency can be calculated according to Equation (3) or expressed according to the relation (*C_x_*/*C*_0_).
(3)$$\%\;photodegradation\;efficiency=\frac{C_{0}-C_{x}}{C_{0}}\times 100$$
where *C*_0_ is the initial methylene blue concentration and *C_x_* is the methylene blue residual concentration in solution after a certain time.

## 4. Conclusions

In this work, results of the synthesis and application of ZnO nanoparticles obtained by an easy-to-perform technique were presented. The nanoparticles synthesized without botanical extract showed catalytic activity, being able to degrade the model dye (methylene blue) in 150 min. When the nanostructures were synthesized in the presence of different botanical extracts, it was determined that the neem leaf (*Azadirachta indica*) and chrysanthemum flower (*Chrysanthemum morifolium*) extracts did not significantly improve the photocatalytic performance of the materials. However, in the presence of shiitake mushroom (*Lentinula edodes*) and cempasuchil flower (*Tagetes erecta*) extracts, it was possible to improve the degradation rate of the pollutant at 120 and 45 min, respectively. These results indicate that, depending on the phytochemicals present in the extracts, it is possible to modify the size of the nanoparticles, as well as affect their adsorption capacity and the generation of oxidizing radicals, which definitely have an impact on the performance of the material for photocatalysis purposes. Further studies using extracts with similar compounds to those of *Tagetes erecta* and *Lentinula edodes* in conjunction with gas chromatography/mass spectroscopy and high-performance liquid chromatography are needed in order to properly design and fully exploit the technological potential of this type of photocatalytic material. Other applications may include those in the biomedical area to inhibit microbial growth and even with potential application to certain cancer types, topics that may be of interest for future research.

## Figures and Tables

**Figure 1 molecules-27-00006-f001:**
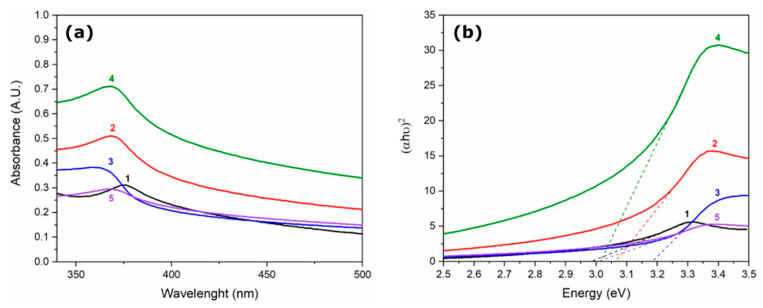
(**a**) Absorption spectra and (**b**) Tauc’s plots for ZnO nanoparticles synthesized with (1 ―) no extract, (2 ―) *Azadirachta indica*, (3 **―**) *Chrysanthemum morifolium*, (4 ―) *Tagetes erecta*, and (5 ―) *Lentinula edodes*.

**Figure 2 molecules-27-00006-f002:**
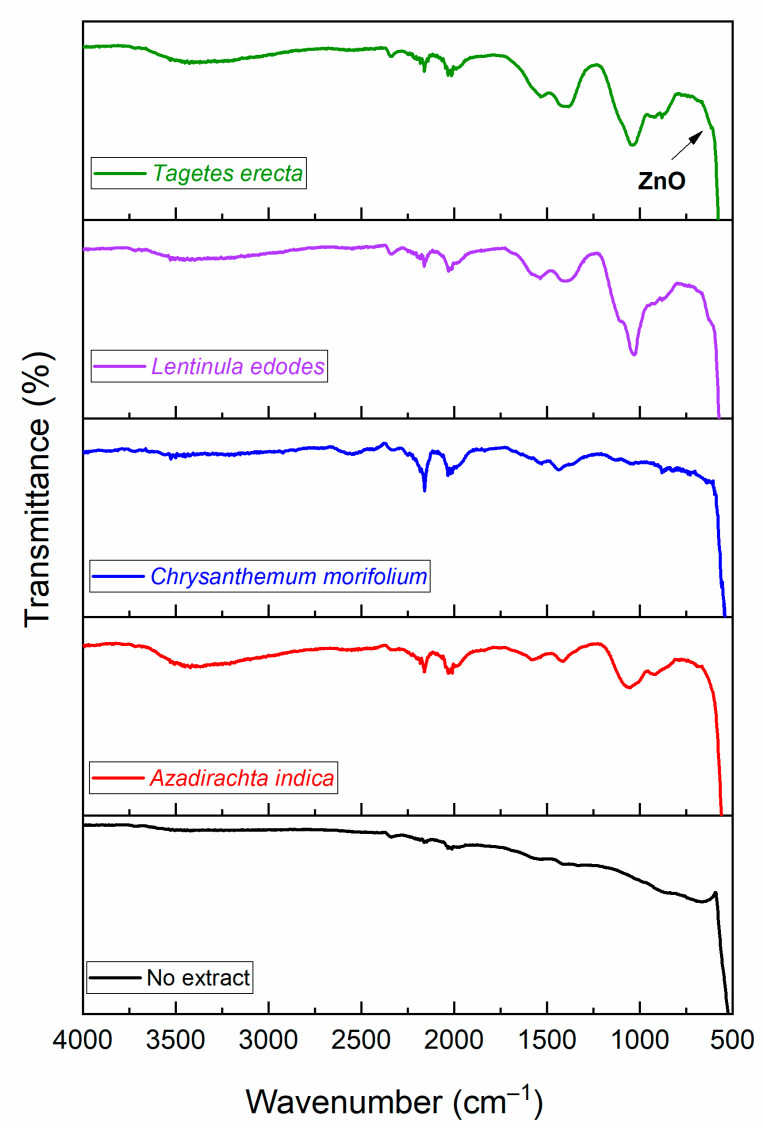
FTIR spectra for ZnO NPs synthesized with different botanical extracts.

**Figure 3 molecules-27-00006-f003:**
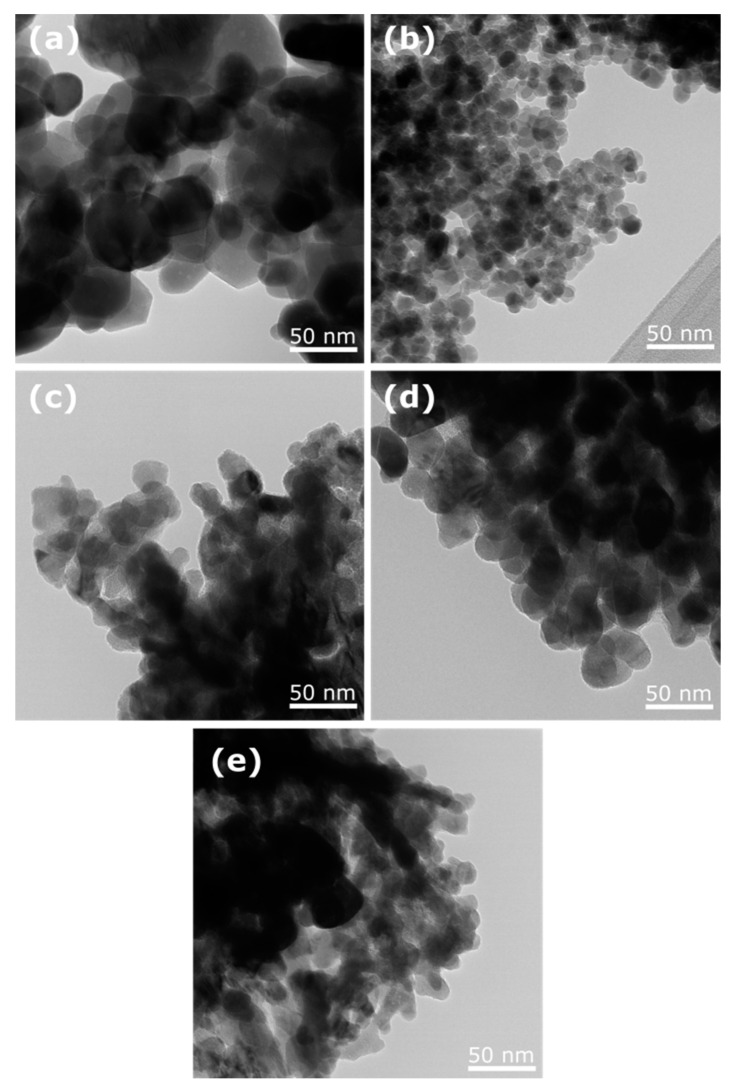
TEM micrographs of ZnO NPs synthesized with (**a**) no extract, (**b**) *Azadirachta indica*, (**c**) *Tagetes erecta*, (**d**) *Chrysanthemum morifolium*, and (**e**) *Lentinula edodes*.

**Figure 4 molecules-27-00006-f004:**
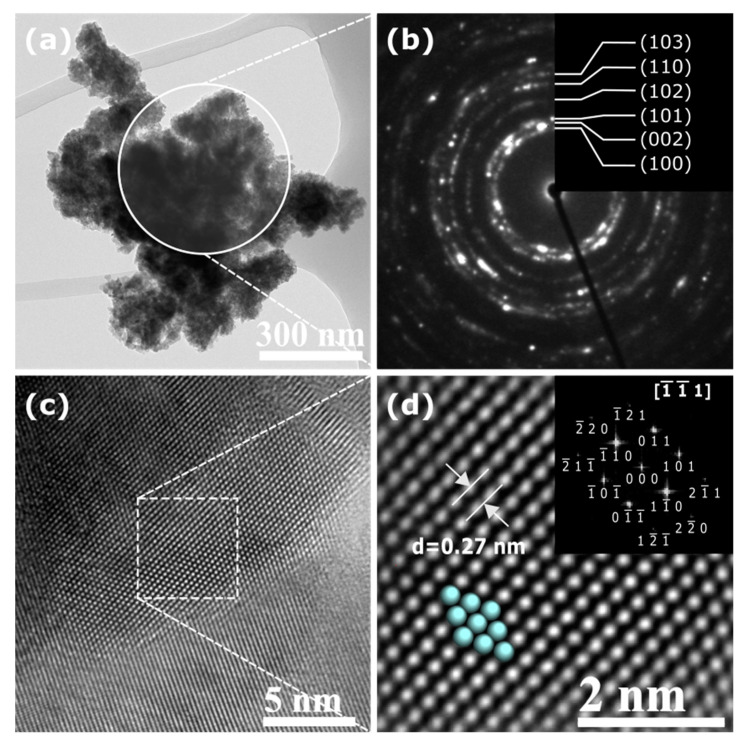
TEM images of ZnO NPs synthesized with *Lentinula edodes*: (**a**) ZnO NPs aggregates, (**b**) indexed SAED pattern, (**c**) HRTEM image of a ZnO nanoparticle, and (**d**) image in (**c**) processed with GMSS together with the FFT pattern.

**Figure 5 molecules-27-00006-f005:**
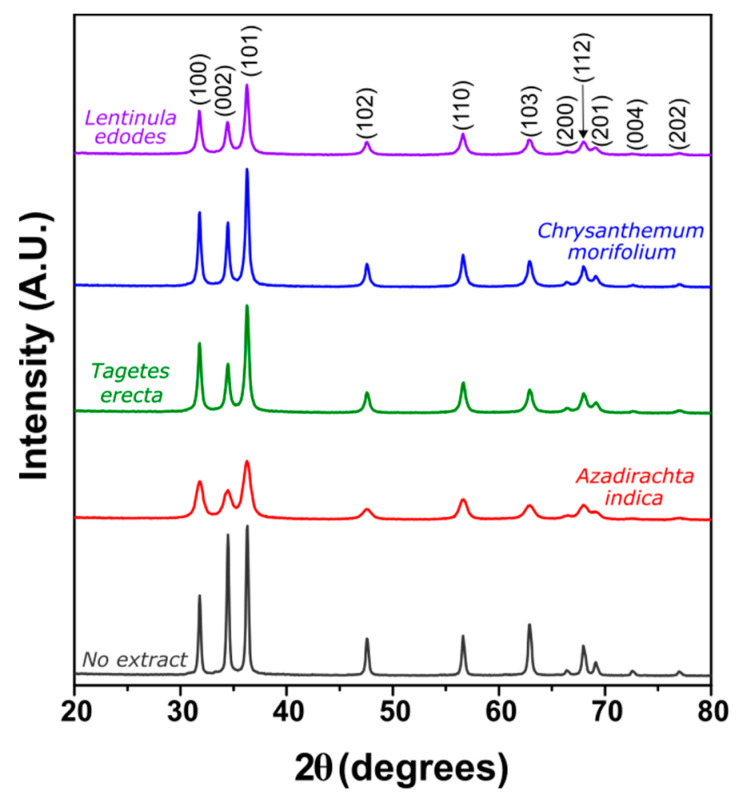
XRD patterns of ZnO NPs synthesized with different botanical extracts.

**Figure 6 molecules-27-00006-f006:**
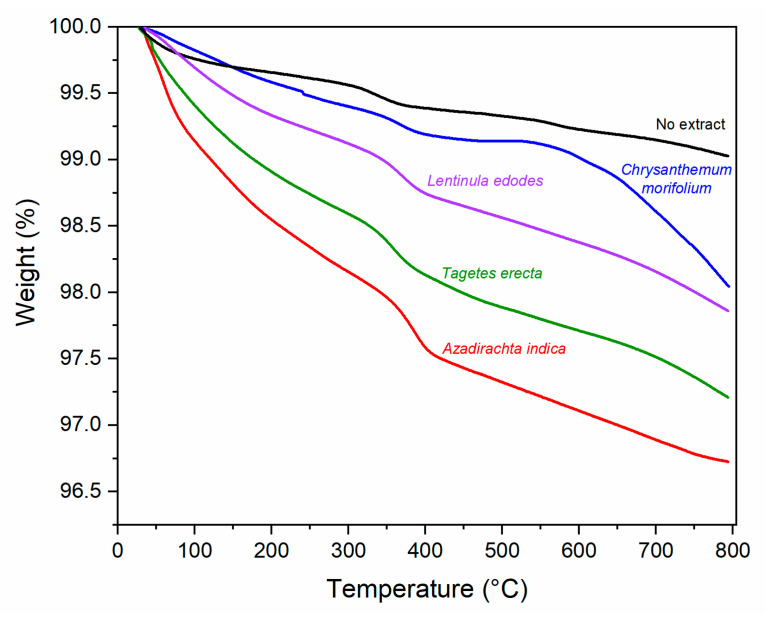
TGA thermograms of ZnO NPs synthesized with no extract and different botanical extracts.

**Figure 7 molecules-27-00006-f007:**
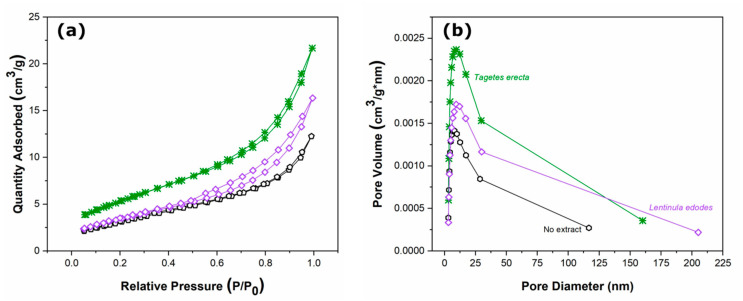
(**a**) Nitrogen adsorption–desorption (BET) isotherms and (**b**) BJH plots of ZnO nanoparticles obtained with (―) no extract, (―) *Tagetes erecta*, and (―) *Lentinula edodes*.

**Figure 8 molecules-27-00006-f008:**
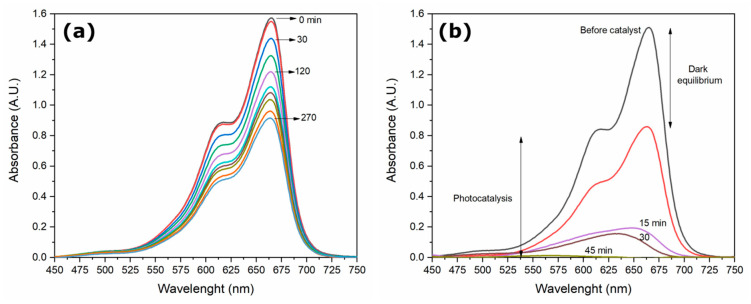
Absorbance spectra of MB: (**a**) no catalyst at 270 min and (**b**) ZnO-*Tagetes erecta* at 45 min.

**Figure 9 molecules-27-00006-f009:**
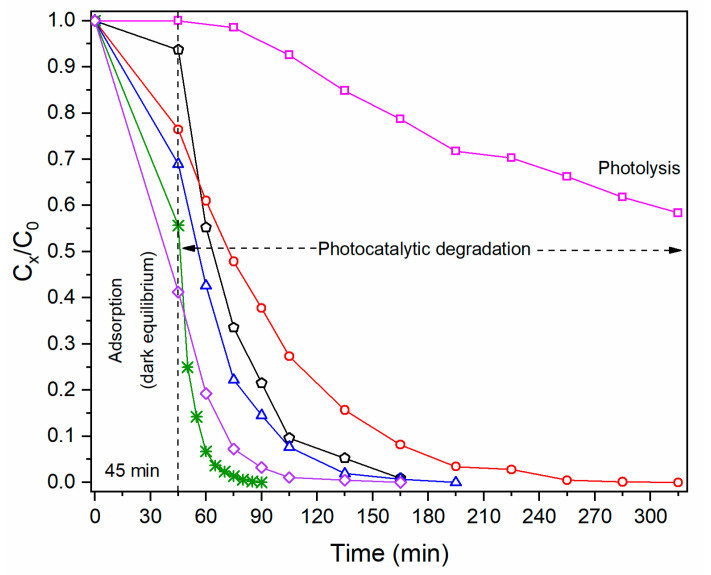
*C_x_*/*C*_0_ plots of MB with (―) No catalyst, (**―**) No extract, (―) *Azadirachta indica*, (―) *Chrysanthemum morifolium*, (―) *Lentinula edodes*, and (―) *Tagetes erecta.* The first 45 min correspond to MB adsorption during dark equilibrium.

**Figure 10 molecules-27-00006-f010:**
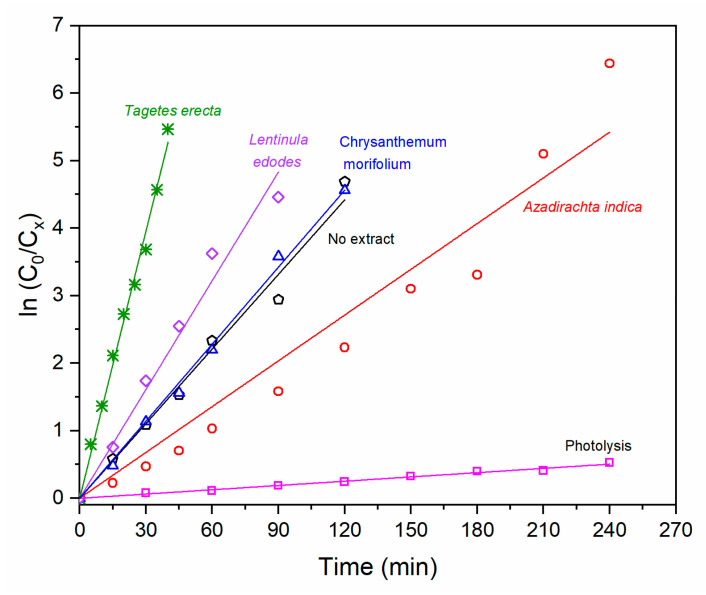
ln(*C*_0_/*C_x_*) as a function of sunlight exposure time of MB and varying catalyst.

**Figure 11 molecules-27-00006-f011:**
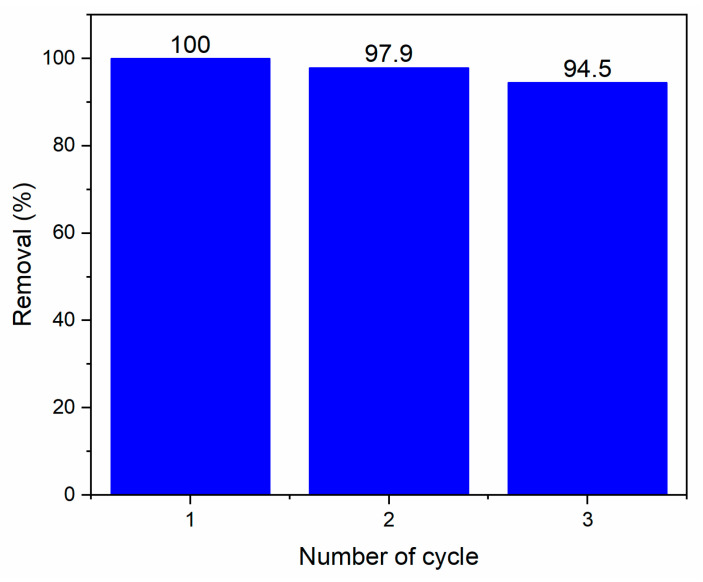
Recycling and reuse of ZnO-*Tagetes erecta* nanoparticles.

**Figure 12 molecules-27-00006-f012:**
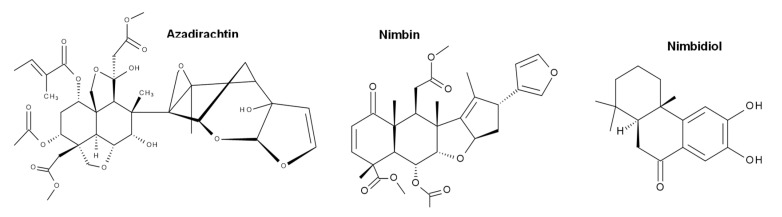
Chemical structure of some of the main components of *Azadirachta indica* extract.

**Figure 13 molecules-27-00006-f013:**
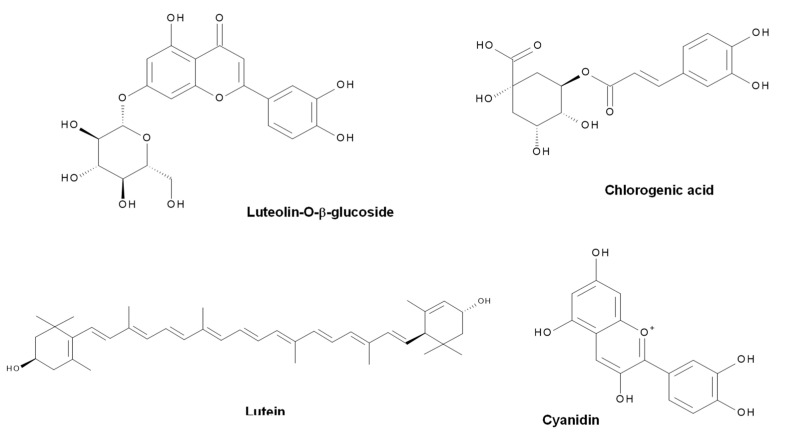
Chemical structure of some of the main components of *Chrysanthemum morifolium* extract.

**Figure 14 molecules-27-00006-f014:**
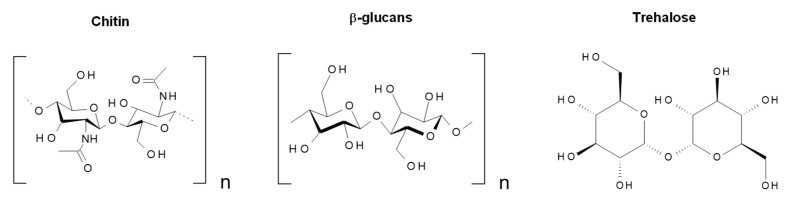
Chemical structure of some of the main components of *Lentinula edodes* extract.

**Figure 15 molecules-27-00006-f015:**
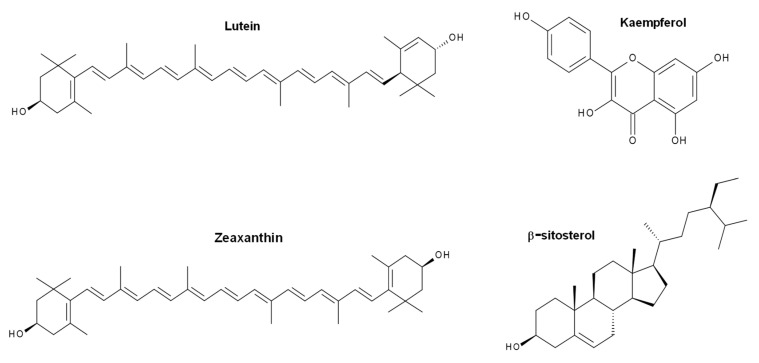
Chemical structure of some of the main components of *Tagetes erecta* extract.

**Table 1 molecules-27-00006-t001:** Size variation of the ZnO NPs.

Extract	Particle Size Variation (nm)	Average (nm)	SD ^1^ (nm)
No extract	12.34–63.29	42.99	16.43
*Azadirachta indica*	9.01–18.97	14.42	3.30
*Tagetes erecta*	24.45–32.15	27.18	2.62
*Chrysanthemum morifolium*	22.92–37.92	31.72	4.42
*Lentinula edodes*	15.52–26.56	20.54	3.70

^1^ Standard deviation.

**Table 2 molecules-27-00006-t002:** Crystallite size of the ZnO NPs.

Extract	Crystallite Size (nm)	SD ^1^ (nm)
No extract	26.53	1.83
*Azadirachta indica*	9.06	1.11
*Tagetes erecta*	16.35	1.30
*Chrysanthemum morifolium*	18.14	1.83
*Lentinula edodes*	14.09	1.26

^1^ Standard deviation.

**Table 3 molecules-27-00006-t003:** BET analysis surface area and porosity of ZnO nanoparticles.

Extract	S_BET_ (m^2^/g)	Pore Volume (cm^3^/g)	Average Pore Diameter (nm)
No extract	11.77	0.0190	6.44
*Tagetes erecta*	19.68	0.0335	6.82
*Lentinula edodes*	12.87	0.0253	7.86

**Table 4 molecules-27-00006-t004:** Apparent rate constant for the photocatalytic degradation of MB dye with ZnO NPs synthesized with different botanical extracts.

System	*k_app_* (min^−1^)	R^2^
No extract (photolysis)	0.00211 ± 4.3524 × 10^−5^	0.99661
ZnO	0.03682 ± 1.1800 × 10^−3^	0.99386
ZnO-*Azadirachta indica*	0.02259 ± 1.1600 × 10^−3^	0.97420
ZnO-*Tagetes erecta*	0.13179 ± 2.0900 × 10^−3^	0.99800
ZnO-*Chrysanthemum morifolium*	0.03802 ± 5.9949 × 10^−4^	0.99851
ZnO-*Lentinula edodes*	0.05369 ± 2.1200 × 10^−3^	0.99224

## Data Availability

Not applicable.

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
