# Peer review of "Sunlight Photocatalytic Performance of ZnO Nanoparticles Synthesized by Green Chemistry Using Different Botanical Extracts and Zinc Acetate as a Precursor"

_molecules, 2021, doi:10.3390/molecules27010006_

Round 1

Reviewer 1 Report

The manuscript by J.R. López et al. entitled Sunlight Photocatalytic Performance of ZnO Nanoparticles Synthesized by Green Chemistry Using Different Botanical Extracts and Zinc Acetate as a Precursor describes the preparation of zinc(II) oxide nanoparticles with the use of plant and fungi extracts. The prepared nanoparticles were then assessed for their ability to remove methylene blue as a model pollutant from a water solution. The article is well written in terms of English and style. It fits the scope of the journal Molecules.

However, there are some major concerns that would have to be addressed before it can be accepted:

  1. Photodegradation: The calculated constants do not refer to photocatalysis but rather to adsorption+photolysis+photocatalysis processes, so basically “MB removal constants”. For the calculation of the photocatalytic degradation constants the given value should be corrected by the adsorption and photolysis. I assume that the C0 mentioned both in Figures 8 and 9 and in the formulas (1) and (2) is the same value, as there is no explanation about it in the text. Also, one thing that is missing is the comparison to non-modified ZnO NPs. Such experiments are not shown and even aren’t discussed once – were the materials better or worse acting on MB than the non-modified ZnO NPs? Another thing - was there an experiment performed in the dark to show the time needed for reaching adsorption-desorption equilibrium or was the 45 minutes of the dark phase chosen arbitrary? My other complain would be that the MB could be removed from the solution by adsorption and not fully degraded, as the authors show high adsorption of MB on the material and assessed the medium absorbance at 665 nm.
  2. From the IR one can clearly see that the materials are doped with the extracts, so their properties rely not only on the morphology changes but also on the presence of the dopants (agglomeration, UV-Vis absorption, charge separation). Because of that the properties of the materials could change over one catalytic experiment. Catalyst recycling should be performed to assure the stability of the materials. Also, why were the UV-Vis absorption spectra recorded only up to 500 nm?
  3. Why did you use such extracts? Was there any specific reason? Currently it seems that the extracts were selected based on their availability to the authors.
  4. UV-Vis, FT IR, XRD, and TGA should be compared to a reference ZnO nanoparticles (prepared without the extracts) to show the effect of the botanical extracts on the nanoparticles. The patterns and spectra presented in the work could use an addition of the data for such reference ZnO NPs.
  5. The names of the sections should be changed/rearranged. The results section contains discussion, so it should be renamed to results and discussion. The current discussion section is a mix between the discussion and conclusions.
  6. Why was indica extract prepared using twice as much of the material per water as compared to other extracts?

Less significant remarks:

  1. Chrysanthemum L. is a genus name, not a species. What species was used?
  2. The part of the sentence “agreeing with the catalysts that adsorbed more MB molecules” (line 23) should be rewritten as it is incomprehensible.
  3. The sentence “MB was completely degraded in 45 min using Tagetes erecta when subjected to solar irradiation, while Lentinula edodes in 120 min.” (lines 23-25) suggests that MB was degraded by the extracts alone and not the materials obtained with the use of the extracts.
  4. There is something off with the pages numbering – they are displayed as “5 of 5” on each page.
  5. “their size and aggregation vary as a function of the extract used” (line 114) – I don’t suppose it is possible to calculate such function. Please rephrase. Also, how did you prove the aggregation of the NPs?
  6. It seems odd that there is still ethanol present in the samples tested in TGA after sintering them at 400°
  7. The paragraph in lines 159-164 contains no reference for the information given.
  8. The value of 0.00211 (line 207) is the constant for “no catalyst” and not indica as stated in the text (see table 2).
  9. Water cannot be refluxed at 70°C using conditions described (line 271).
  10. English/style/typos: “TiO2” – subscript (line 39); “hibiscus” should be capitalized (62); the Latin names should be italicized (63, 85-86, 116, 117, 140, 142, 153, 214-215); “ultrasound” – ultrasonic bath? (269); “were dissolved” – was mixed (267).

Author Response

Thank you for reviewing our work and for the comments that helped us improve the manuscript.  Responses to your requests are in the attached document.

Reviewer 2 Report

The authors prepared a series of ZnO NPs by green chemistry using different botanical extracts and zinc acetate as a precursor and investigated their sunlight photocatalytic performances. The photocatalytic activity is great depended on the composition of the extract used, the size and porosity of ZnO NPs. The results are very interesting and recommended to be published on Molecules after major revision. The following aspects need to be considered for improvement.

  1. The structure and the photocatalytic performances of ZnO NPs that prepared without using botanical extracts must be presented for better understanding the function of botanical extracts.
  2. The catalytic properties of ZnO NPs is related to their porosity. Therefore, the N2-physisorption of ZnO NPs should be carried out.
  3. The structural differences of botanical extracts should be presented in the manuscript.
  4. Please calculate the particle size of ZnO NPs by the Scherrer Equation.

Author Response

(The authors gave the same response as above.)

Round 2

Reviewer 1 Report

The manuscript improved considerably after the corrections provided by authors and can be accepted in its revised version.

Reviewer 2 Report

After the modifications, this manuscript might be accepted for the publication.